Yoga pose recognition using dual structure convolutional neural network

Meng Xiang xiangmenghn@163.com
Liu Zhaobing
Hunan University of Medicine , Hunan , China
Coelho Paulo Jorge
Electronic publication date: 2025 May 27
Publication date: 2025
Volume: 11
Electronic Location ID: e2907
Received 2024 Jul 24; Accepted 2025 Apr 28
Copyright: © 2025 Meng and Liu
Copyright year: 2025
Copyright holder: Meng and Liu
License: This is an open access article distributed under the terms of the Creative Commons Attribution License, which permits unrestricted use, distribution, reproduction and adaptation in any medium and for any purpose provided that it is properly attributed. For attribution, the original author(s), title, publication source (PeerJ Computer Science) and either DOI or URL of the article must be cited.
License URL: https://creativecommons.org/licenses/by/4.0/

Keywords: Convolutional neural networks, Feature fusion, Yoga pose

Funding: Hunan Provincial Department of Education 22C1191 Design of Characteristic Physical Education Options for Female Students in Medical Colleges Hunan University of Medicine 2021JG24 This work was supported by the Study on the efficacy of yoga intervention in primary dysmenorrhea, 2022 scientific research project of Hunan Provincial Department of Education, No. 22C1191 and the research on the Design of Characteristic Physical Education Options for Female Students in Medical Colleges, 2021 Teaching Reform Project of Hunan University of Medicine, No. 2021JG24. The funders had no role in study design, data collection and analysis, decision to publish, or preparation of the manuscript.

==============================
As a popular form of physical and mental exercise, the correct execution of yoga movements is crucial. With the development of deep learning technologies, automatic recognition of yoga postures has become popular. To recognize five different yoga postures, this article proposed a dual structure convolutional neural network with a feature fusion function, which consists of the convolutional neural network A (CNN A) and convolutional neural network B (CNN B). Among them, the structure CNN A observes different channels finding the global feature of yoga images, and the structure CNN B calculates the depth information in each pixel of the yoga images. Following that, the extracted global feature and local feature are fused by a feature fusion function of taking a matrix dot multiplication. Finally, the softmax layer accurately recognizes yoga postures based on the fused features. Experimental results show that the proposed model achieves 97.23% accuracy with 96.08% precision and defeats against the competitors in the recognition of yoga postures. Moreover, the feature fusion function is proved to be successful in terms of the recognition to yoga postures. We also find that the feature fusion with a matrix dot multiplication operation can significantly improve the recognition accuracy of yoga postures than that with a direct connection operation.

Introduction

With the improvement of health awareness, yoga movements, which are treated as an effective means of physical and mental practice, are becoming increasingly popular among people. Yoga not only improves the flexibility of the body and balances the body, but also reduces stress and improves mental concentration. However, the correct yoga postures are the key to practices yoga, and incorrect postures can lead to injury. Recently, using machine learning and deep learning techniques for yoga pose recognition and correction has become a hot research topic. With the advancement of techniques and the enrichment of datasets, the application prospects of machine learning in yoga pose recognition are very broad. For those approaches based on neural network structures, they can also exhibit good accuracy and efficiency in yoga pose recognition. Additionally, with the popularity of wearable devices, combining machine learning techniques with these devices will make real-time monitoring and correction of yoga pose feasible and convenient.

Deep learning techniques have been widely applied in yoga pose recognition due to its outstanding performance in image recognition and processing (Anwarul & Mohan, 2022). Identifying different yoga postures through training algorithms can help yoga practitioners to improve their postures, thereby achieving better exercise results and avoiding harm. Convolutional neural networks (CNNs) are one of the most used deep learning models for image recognition tasks. Researchers have developed various CNNs based models to recognize yoga postures. These models can identify specific yoga movements and provide improvement suggestions by analyzing the images of yoga practitioners.

For example, Talaat (2023) proposed a novel deep learning model for estimating yoga postures. The model identifies and corrects incorrect postures by analyzing the yoga pose attributes in images. Ashraf et al. (2023) designed a neural network model called YoNet for classifying yoga postures. The research focuses on how to use deep learning techniques to improve the accuracy of pose recognition and classification. Garg, Saxena & Gupta (2023) employed a deep learning architecture that combines convolutional neural networks and media pipe technology for yoga pose recognition. Prasiddha Sarma (2024) used the weight encoder-decoder to recognize yoga postures, and obtained satisfactory results. Chaudhari et al. (2021) described a real-time yoga pose correction system, namely Yog Guru based on deep learning methods, which can provide real-time feedback on the practitioner’s pose.

Motivation

To address the recognition of yoga postures, this work designed a dual structure convolutional neural network with a feature fusion function. On the one hand, we sufficiently borrow the ascendency of convolutional neural networks for feature extraction on images. With the assistance of this ascendency, the global feature and the local feature of a yoga image can be extracted. Hence, we take a dual structure convolutional neural network consisting of the structure CNN A (convolutional neural network A) and the structure CNN B (convolutional neural network B). On the other hand, using the proposed feature fusion function to fuse the extracted global feature and local feature, by doing so, the details at different levels of a yoga image can be focused on, which is beneficial for the recognition of yoga poses.

Contributions

The contributions of this work are summarized. (1) A dual structure convolutional neural network is used for the recognition of yoga postures, where the structure CNN A can observe different channels of finding the global features, and the structure CNN B can calculate the depth information in each pixel of an image. Together, the two structures pay attention to critical features of yoga images.

(2) The proposed feature fusion function is proved to have positive effects on the recognition of yoga postures. Moreover, the feature fusion with a matrix dot multiplication operation can yield more effectiveness in the improvement of the recognition for yoga postures than that with a direct connection operation does. The values of feature fusion can afford the direction of ongoing yoga and human posture.

This work thus is arranged below as follows. The following section, the related literature is summarized in Related Works. Methodology discusses the proposed model architecture and the proposed feature fusion function. Experiments and Results puts a discussion based on the experimental analytics. Lastly, Conclusion draws this work and directs future works.

Related works

Chen, Tian & He (2020) and Jose & Shailesh (2021) focus on body pose estimation of image recognition and summarize the ascendency of deep learning methods in the recognition of body postures. Deep learning techniques, particularly CNNs and their variants, have significantly improved the accuracy, robustness, and efficiency of recognizing and tracking human body postures from images or videos. Deep learning models have surpassed traditional computer vision methods in accurately identifying key points of the human body, such as joints and limbs. This enables precise estimation of body postures even in challenging conditions like occlusions or varied backgrounds (Pala et al., 2023). For example, CNN architectures can handle complex poses involving multiple individuals or intricate body configurations through introducing techniques like residual connections and attention mechanisms. Huang et al. (2021) developed a yoga training system based on real-time pose estimation, which consists of a posture recognition network, a yoga standard movement posture library, a yoga movement correction algorithm and a system UI interface. To address the issue of emotional oversight in yoga practices, Duppala et al. (2024) presented an innovative approach to personalized yoga practice by leveraging deep learning and computer vision techniques for real-time monitoring and correction of yoga poses. Kulkarni et al. (2024) utilized convolutional neural networks and long short-term memory to precisely recognize and continuously monitor yoga poses, however, the model needs sufficiently training sets to ensure the accuracy. Similarly, Lavanya, Kalila & Rohith (2024) and Anwarul & Mohan (2022) utilized a deep learning method to recognize yoga poses, and obtained satisfactory recognition accuracy. Pavikars (2024) used a convolutional neural network to estimate yoga poses. Shirisha et al. (2024) used VGG 16 to handle yoga pose recognition and obtained superior recognized results. Overall, deep learning methods have ushered in a new era of precision and scalability in the recognition of body poses. Their ability of handling complex and varied pose scenarios has opened numerous applications across industries, driving advancements in healthcare, sports analytics, human-computer interaction, and beyond. Continued innovation in deep learning algorithms promises further enhancements in pose estimation accuracy and real-time performance, paving the way for more intelligent and responsive systems.

Additionally, aiming for recognition tasks of yoga postures, Agrawal, Shah & Sharma (2020) conducted multiple experiments by machine learning techniques. Through implementing multiple experiments on 5,500 images containing ten different yoga postures, Agrawal, Shah & Sharma (2020) find that these classifiers based on random forest can work better on the experimental dataset. Liaqat et al. (2021) combined traditional machine learning approaches with deep neural networks for yoga pose recognition, and satisfactory results are achieved in identifying different yoga postures. Kumar & Sinha (2020) used the OpenPose to build the model for six yoga poses in 88 videos. Although the built model is time consuming for identifying the six yoga poses, the built model can accurately identify different yoga poses. In terms of machine learning techniques, the challenges of identifying yoga postures still exists, such as handling occlusions, pose ambiguity, and the need for annotated datasets for training robust models (Raza et al., 2023; Bera et al., 2023; Rajendran & Sethuraman, 2023).

Methods

This section is to fulfill feature information integration to recognize yoga poses. Specifically, we took into account integrating critical feature information of yoga poses and then using the integrated feature information to identify yoga poses. Accordingly, a dual structure convolutional neural network, namely DSCNN, is proposed, where one convolutional neural network is used for global feature extraction and the other is used to focus on the local feature. Thereafter, the characteristics extracted by the two convolutional neural networks are fused by a feature fusion function. The details are as follows:

Model’s structure

Figure 1 displays the structure of DSCNN, including the structure CNN A and the structure CNN B. Because the two structures CNN A and CCN B focus on different objectives, CCN A adopts the two-layer structure, while CNN B adopts the three-layer structure. The details are as follows.

Figure 1 Architecture of DSCNN.

The structure CNN A is used to focus on the global feature of a yoga image. To extract the global feature at coarse granularity, in structure CNN A, we consider a double layer structure, including 50×50×16 convolution layer, batch normalization and rectified linear unit (ReLU), followed by 25×25×128 convolution layer and a MaxPooling layer.

The structure CNN B is to finely extract the local feature of a yoga image, which can pay more attention to the details of a yoga pose, i.e., the so-called ‘local feature’. In structure CNN B, we designed three convolutional layers, including 50×50×32 convolution layer, batch normalization and ReLU, followed by 50×50×64 convolution layer, batch normalization, ReLU and a MaxPooling layer. Thereafter, 25×25×128 convolution layer, batch normalization ReLU, and a MaxPooling layer. After the global feature and the local feature are integrated, they enter a fully connected layer. Finally, the recognition of yoga poses is implemented in the Softmax layer.

Here, the global feature extraction and the local feature extraction have different purposes. For example, like the two yoga poses in the literature (Fig. 5C in Verma et al., 2020), when the two yoga poses are the same, but there are obvious differences in the background, to prevent misrecognition caused by background interference, hence, we considered the global feature. That is, the role of the global feature is to resist the interference of the background in an image. However, when the two yoga poses have obvious differences, in this scenario, the local feature can directly recognize yoga poses. Although our goal is to recognize yoga poses of input images, the background of input images may create interference. Accordingly, we took into account the global feature to resist the interference. Certainly, the recognition of yoga poses mainly relies on the local feature. Overall, the roles of the local feature and the global feature are different.

Feature fusion

This subsection starts to illustrate how to fuse the global feature and the local feature. For the convenience of description, let us assume the global feature Fg=[g11g12...g1mg21g22...g2m............gm1gm2...gmm] and the local feature Fl=[l11l12...l1ml21l22...l2m............lm1lm2...lmm].

The feature fusion function Ω is given in Eq. (1).

(1) Ω=(Fg∗α)⊗(Fl∗(1−α))

where symbol ⊗ represents a matrix dot multiplication operation. Item α is a weight coefficient and the calculation is below. The mean of Fg and Fl is calculated as follows,

(2) {u(Fg)=1m∑i,jmgiju(Fl)=1m∑i,jmlij.

The corresponding standard variance is calculated in Eq. (3)

(3) {σ(Fg)=(1m∑i,jm(gij−u(Fg))2)12σ(Fl)=(1m∑i,jm(lij−u(Fl))2)12.

Now, let us calculate the item α. As follows

(4) α={β1,ifβ1<β2β2,ifβ2<β1

(5) {β1=Δ1Δ1 + Δ2β2=Δ2Δ1 + Δ2Δ1=u(Fg)|σ2(Fg) +(1m∑i,jm(gij−u(Fg))3)13|12Δ2=u(Fl)|σ2(Fl) +(1m∑i,jm(lij−u(Fl))3)13|12

where items (1m∑i,jm(gij−u(Fg))3)13 and (1m∑i,jm(lij−u(Fl))3)13 are to increase the weight ratio between the global feature and the local feature, thus allowing the local feature to obtain higher weights.

Illustration

We utilize the mean u(Fg), u(Fl) in Eq. (2) and the standard deviation σ(Fg), σ(Fl) in Eq. (3) to calculate weight coefficient α in the feature fusion function Eq. (1). This is to effectively integrate the global feature and the local feature from a distribution perspective.

Model parameters

Convolutional kernel. The main purpose of the convolution operation is to achieve feature extraction of an image. The Convolutional kernel includes High Pass Filter (HPF) and Low Pass Filter (LPF), among them, HPF means that the high-frequency part of an image (i.e., the part with more dramatic changes in an image) is allowed to pass through. HPF is often used to sharpen an image and enhance the edges of objects in the image. Such as Sobel operator, Prewitt operator, sharpening filter, etc. However, LPF implies that the low-frequency part of an image (i.e., the part with the slowest change in an image) is allowed to pass through. LPF is often used to blur or smooth an image, eliminate noise, etc. Such as Gaussian filter, mean filter, etc. In terms of our DSCNN, we chose a 3 × 3 4-adjacent Laplace operator. Because the edges in an image are those areas where the grayscale changes, the Laplace operator is very useful in edge detection.

Max pooling. During the pooling process, max pooling only retains the maximum value of each region, ensuring that the strongest features can be retained and other weak features are discarded. This can assist the improvement of recognition ability for the model to key features. Therefore, a 2 × 2 max-pooling is used in our DSCNN.

Batch size. Figure 2 displays the relationship between batch size, training epoch and training accuracy (Smith et al., 2017). It can be seen that the processing speed to the same amount of the data augments with the increase of batch size. Due to the final convergence accuracy falls into different local extremes, the batch size is increased to a certain size to obtain the best final convergence accuracy. Taking Fig. 2 as a reference, the batch size is dynamically adjusted during the training phase.

Activation function. The most commonly used ReLU activation function is used in our DSCNN. Equation (6) gives the ReLU activation function

Figure 2 Batch size.

The results are cited in Smith et al. (2017).

(6) f(x)=max(0,x)

The reviewer ReLU function is a piecewise linear function, which sets all negative values to 0, while it leaves positive values unchanged. This operation is called lateral inhibition, because of the lateral inhibition, neurons in neural networks exhibit sparse activation. This is particularly evident in deep neural network models, such as CNNs. When training a deep classification model, there are often only a few features related to the target. Indeed, a sparse model achieved through ReLU can better mine relevant features and fit the training data. Consequently, we chose the ReLU function as our activation function.

Algorithm implementation

The algorithm is as shown in Algorithm 1. The input and output are an image set Iset, training accuracy and testing accuracy, respectively. Firstly, image set Iset is randomly divided into the training set Trainset and testing set Testset, where 70% of the image set is used for the training set, and the rest is used as the testing set, illustrated in Step 1 to Step 3. Following that, the structures CNN A and CNN B are initialized in Step 4 and Step 5. The process between Step 6 and Step 21 describes model’s training. Structure CNN A extracts global features Fg and Structure CNN B extracts local features Fl, illustrated in Step 7 to Step 10. The extracted global features and local features are fused by the feature fusion function Eq. (1), as shown in Step 11 to Step 14. According to the fused features, Softmax layer recognizes yoga postures in Step 15. If the model starts to converge, the training is terminated. Current training model is saved, and the training accuracy is outputted, as shown in Step 16 to Step 20. Finally, using the testing set Testset to test the trained model, and the testing accuracy is outputted, illustrated in Step 22 and Step 23.

Algorithm 1 Input     image set Iset	
Output    training accuracy, testing accuracy	
1         Dividing Iset into training set Trainset and testing set Testset;	
2         Trainset = 70%* Iset ;	
3         Testset = Iset - Trainset;	
4         Initializing structure CNN A;	
5         Initializing structure CNN B;	
6         For j = 1 to Jmax:	
7            Using Trainset to train CNN A;	
8            Extracting global features Fg;	
9            Using Trainset to train CNN B;	
10           Extracting local features Fl;	
11           Using Eqs. (2) and (3) to calculate u(Fg), u(Fl), σ(Fg), σ(Fl);	
12           Using Eqs. (4) and (5) to calculate α;	
13           Using feature fusion function Eq. (1) to fuse Fg and Fl;	
14           Obtaining the fused features Λg⊗Λl;	
15           Softmax layer recognizes yoga poses by Λg⊗Λl;	
16           If the training converges then:	
17             Saving current training model;	
18             Output training accuracy;	
19             break;	
20           End if	
21        End for	
22        Using Testset to test the training model;	
23        Output testing accuracy;	

Experiments and results

Experimental datasets

The experimental dataset is cited from the Yoga-82 dataset in Verma et al. (2020). We chose five types of yoga poses from the Yoga-82 dataset, including Standing yoga pose, Twisting yoga pose, Forward Bend yoga pose, Kundalini yoga pose and Prenatal yoga pose. (Regarding the four yoga pose and the Yoga-82 dataset, please refer https://sites.google.com/view/yoga-82/home). The experimental dataset consists of 280 images, of which 70% is used for the training set, and the rest 30% for validation set.

To evaluate our DSCNN, we selected five competitors Resnet 50 (He et al., 2016), Inception V3 (Szegedy et al., 2016), Xception (Chollet, 2017), Inception-Resnet-V2 (Szegedy et al., 2017), YoNet (Ashraf et al., 2023). Then, using the four metrics Accuracy, Precision, Recall, F1-score to evaluate their recognition ability to yoga poses. Unless otherwise stated, our model and the five competitors are running in the same experimental environment.

Experimental designs

To verify the proposed method, multiple experiments were designed. (i) To verify the effectiveness of the feature fusion function, the ablation experiments were carried out. Specifically, four manners were designed to recognize the five types of yoga poses. The first one is that we singly use the structure CNN A in Fig. 1 to recognize yoga poses, and in the second manner, using the structure CNN B in Fig. 1 to identify yoga poses. As for the third manner, denoted as CNN A + CNN B, the local feature extracted by CNN B and the global feature extracted by CNN A are directly connected, rather than using the feature fusion function Ω, i.e., the global feature is directly followed by the local feature. While for the fourth manner, using the feature fusion function Ω, i.e., using our DSCNN to recognize the five types of yoga poses.

(ii) To verify recognition performance, we compared the DSCNN and the five competitors Resnet 50, Inception V3, Xception, Inception-Resnet-V2, YoNet. Using the four evaluated metrics Accuracy, Precision, Recall, F1-score to analyze the recognition results.

Result analysis

This subsection displayed the experimental results, which contains the validation on feature fusion and the recognition ability to the five types of yoga poses. Experimental results imply that our DSCNN defeated against the five competitors in recognition performance to the five types of yoga poses, and achieved 97.23% accuracy with 96.08% precision, The detailed results are illustrated as follows.

(i) Validation on feature fusion

To verify the feature fusion function Ω in Eq. (1), the ablation experiment was designed. The results of the ablation experiment are illustrated in Table 1. Through observing the four evaluation metrics accuracy, precision, recall, F1-score, we find that the best results are obtained by DSCNN, i.e., the fourth manner. The won advantages of DSCNN imply that the feature fusion function Ω successfully fuses the extracted global features and the extracted local features, so that the recognition performance of the DSCNN is significantly augmented. Together, these results in the ablation experiment confirm that the proposed feature fusion function can assist the improvement of the recognition ability of the model.

Table 1 Results on ablation experiment.

Manner	Models	Illustrations	Accuracy	Precision	Recall	F1-score	
The fourth	DSCNN	Using Ω and feature fusion	97.23%	96.08%	96.62%	96.35%	
The third	CNN A + CNN B	Without using Ω and feature direct connection	91.15%	90.37%	91.27%	91.10%	
The second	CNN B	Without using Ω	87.89%	88.42%	85.52%	84.04%	
The first	CNN A	Without using Ω	83.07%	80.33%	82.30%	82.16%	

(ii) Comparisons of recognition performance

This subsection compared our DSCNN against the five competitors Resnet 50, Inception V3, Xception, Inception-Resnet-V2, YoNet in terms of the recognition of the five types of yoga poses. Compared results in Table 2 show that our DSCNN defeats against the five competitors. In terms of the evaluated metric accuracy, our DSCNN is 97.23%, which is superior to the five competitors. However, the competitor Inception-Resnet-V2 obtains the lowest accuracy. Observing the three evaluation metrics precision, recall, F1-score, our DSCNN also shows significant advantages.

Table 2 Recognition results of different models.

Models	Accuracy	Precision	Recall	F1-score	Average ranks	p = 0.05	
DSCNN	97.23%	96.08%	96.62%	96.35%	1.671	*	
Resnet 50	91.55%	91.80%	91.55%	87.11%	1.803	*	
Inception V3	86.47%	90.03%	86.39%	87.16%	1.855	*	
Xception	89.89%	90.22%	89.82%	89.74%	1.698	*	
Inception-Resnet-V2	81.29%	81.62%	81.35%	81.27%	1.817	*	
YoNet	94.91%	95.62%	94.92%	94.90%	1.778	*	
Note:

Average ranks are given in the right column. The sign ‘*’ shows significant at p = 0.05 level.

To test the statistical significance of the difference between the six algorithms, the Wilcoxon-test was adopted. Average ranks of the six algorithms are calculated by using (∑i=16rij)/6, where rij is the ranking of i-th algorithm on j-th dataset. The tested results in Table 2 show that our DSCNN obtaining the best average ranks is statistically better than the five competitors at the 95% confidence level. Furthermore, there are no differences between them for these comparative results.

Additionally, the fundamental target of our DSCNN can fuse the global feature and the local feature of an image to implement a recognition decision. For that purpose, we explore the feature map for both merged types of convolutions and visualize them, illustrated in Fig. 3. For the yoga poses—“Twisting”, Fig. 3A displays the feature map in the third convolutional layer of the structure CNN A where we can observe different channels finding the global features, such as edges. Figure 3B displays the feature map in the third convolutional layer of the structure CNN B that calculates the depth information in each pixel of the image. Figure 3C displays the fused feature map in the third convolutional layer of our DSCNN.

Figure 3 Feature map corresponding to “Twisting” pose.

(A), (B) Displays the feature map in the third convolution of structure CNN A and of structure CNN B, respectively. (C) Displays the feature map in the third convolution of DSCNN. The details of (A)–(C) is displayed by local amplification.

To further analyze the recognized results, Fig. 4 displays the confusion matrix, showing that the recognition accuracy for the Prenatal yoga poses is lower, while the recognition accuracy on other yoga poses is higher. Additionally, Fig. 5 displays the loss in process of training and validation. Through observation, our DSCNN can converge when the training reaches 110 epochs, and the loss value is 0.44. Moreover, there was no significant vibration in the loss value during the training process, indicating that our DSCNN has certain robustness.

Figure 4 Confusion matrix.

Labels 0–4 are Standing yoga pose, Twisting yoga pose, Forward Bend yoga pose, Kundalini yoga pose and Prenatal yoga pose in order.

Figure 5 Loss values.

Discussion

Advantages

The proposed method outperforms the comparative methods, and the reason can be interpreted as follows. To accurately recognize the five different yoga poses, we considered the global features and local features. This is to pay attention to the five postures from different perspectives. Then, using the feature fusion function Eq. (1) to fuse the extracted global features and local features. In fact, the feature fusion function Eq. (1) is a key factor in our success since it assigns higher weights to the local feature, thereby allowing our model to pay attention to the differences in the five different yoga postures. That is why the proposed method won over the competitors.

Limitations

For the recognition of yoga postures, our method relied on the feature fusion function Eq. (1). Together, the quality of both the extracted global features and local features determined the calculation accuracy of feature fusion function Eq. (1). Here, this work did not consider noisy interference, that is to say, the anti-noisy ability of the feature function Eq. (1) is not proven. If the local features of yoga postures are masked by noise, the proposed model might obtain a poor recognition accuracy. This does not mean that the proposed model does not have the ability to resist noise. The calculation accuracy of the feature fusion function Eq. (1) may decrease, leading to a decline in our model’s ability to recognize yoga postures.

Insights

However, in practical applications, yoga pose recognition faces various challenges. (i) Limitations of the dataset. Yoga pose recognition relies on a large amount of annotated data for model training. But unfortunately, existing public datasets often lack diversity and cannot cover all types of yoga poses and different practitioner states. Especially for some complex yoga poses, the sample size in the dataset may be very limited, which limits the model’s generalization ability. (ii) Real time and accuracy. In practical applications, yoga pose recognition requires real-time operation on mobile devices, which places high demands on the computational efficiency of the algorithm. Meanwhile, to ensure the accuracy of recognition, the algorithm needs to be able to handle various lighting conditions, background interference, and changes in human posture. These factors will all affect the accuracy and real-time performance of recognition. (iii) Detection of joints and key points. The core of yoga pose recognition is the detection and localization of key points on the human body. However, the posture changes of human joints are diverse, sometimes the joints may be obstructed, or in some poses, the posture of the joints may be very close, which increases the difficulty of key point detection. In addition, differences between individuals can also affect the accuracy of key point detection.

Conclusion

To accurately recognize yoga poses, this work proposed a dual structure convolutional neural network with a feature fusion function, which consists of the structure CNN A and the structure CNN B. On the one hand, to extract the global feature of a yoga image, the structure CNN A containing two convolutional layers is designed. On the other hand, to extract the local feature of the yoga image, we designed the structure CNN B containing three convolutional layers. Then, a feature fusion function taking a matrix dot multiplication is used to fuse the extracted global feature and local feature. Finally, experimental results show that the proposed model outperforms the competitors in the recognition of yoga poses. Results also imply that the feature fusion function has positive effects on the recognition of yoga poses. The feature fusion with a matrix dot multiplication operation has more effectiveness in the improvement of the recognition to yoga poses than that with a direct connection operation does. In future work, we will utilize graph neural networks to recognize different yoga poses. The values of this work add suggestions to ongoing yoga and human posture detection and provide future directions for researchers in this field.

Supplemental Information

Supplemental Information 1 Code.

Supplemental Information 2 Dataset 1.

Supplemental Information 3 Dataset 2.

Supplemental Information 4 Dataset 3.

Supplemental Information 5 Dataset 4.

Supplemental Information 6 Dataset 5.

Supplemental Information 7 Dataset 6.

Additional Information and Declarations

Competing Interests

The authors declare that they have no competing interests.

Author Contributions

Xiang Meng conceived and designed the experiments, performed the experiments, analyzed the data, performed the computation work, prepared figures and/or tables, authored or reviewed drafts of the article, and approved the final draft.

Zhaobing Liu analyzed the data, performed the computation work, prepared figures and/or tables, and approved the final draft.

Data Availability

The following information was supplied regarding data availability:

The raw data are available in the Supplemental Files and available at https://sites.google.com/view/yoga-82/home.

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
