# Peer review of "Yoga pose recognition using dual structure convolutional neural network"

_PeerJ Computer Science, doi:10.7717/peerj-cs.2907_

## Round 0.1 · original submission · Major Revisions

After carefully considering the reviews and assessing your manuscript, I am pleased to inform you that we would like to invite you to revise and resubmit your manuscript for further consideration. The reviewers have provided constructive comments that will help strengthen your work. Please address each of these points thoroughly in your revised manuscript. Additionally, ensure that you provide a detailed response letter outlining how you have addressed each comment raised by the reviewers. This will help the reviewers and myself to evaluate the changes made to the manuscript.

Reviewer 1 ·

Basic reporting

The manuscript, while generally clear, suffers from several issues that need addressing. The English language, though mostly professional, contains awkward phrasing and grammatical errors that can hinder comprehension, requiring a thorough linguistic revision. The literature review, although adequate, could benefit from more critical engagement with recent studies to strengthen the context of the research. The article structure, though conforming to basic standards, presents figures that are not as informative as they could be, particularly Figure 1, which lacks sufficient detail to fully explain the DSCNN architecture. Additionally, while the results are presented clearly, there is a lack of depth in the analysis, particularly in discussing the implications of the findings in relation to existing work. Finally, the manuscript would benefit from more rigorous proofreading to ensure that all technical terms are accurately defined and consistently used throughout.

Experimental design

The manuscript presents original primary research that aligns with the Aims and Scope of the journal, but there are significant areas that require improvement. While the research question is defined and relevant, the explanation of how the study fills an identified knowledge gap is somewhat superficial. The manuscript could benefit from a more thorough discussion of the novelty of the approach compared to existing methodologies in yoga pose recognition.

The investigation, though generally rigorous, lacks a detailed description of the experimental setup and ethical considerations, particularly in relation to data collection and participant privacy (if applicable). The methods section, while providing a basic outline of the dual structure convolutional neural network (DSCNN), falls short in offering sufficient detail for full replication of the study. Key aspects, such as the rationale behind the selection of specific hyperparameters and the choice of certain dataset partitions, are not adequately justified. Providing a more comprehensive explanation of the methodological choices would significantly enhance the replicability and reliability of the research.

Validity of the findings

The manuscript presents findings that are generally valid; however, there are several areas where the presentation of these findings could be improved to meet higher standards. The impact and novelty of the research are not explicitly assessed, which is a significant oversight given the importance of these factors in evaluating the contribution of the study to the field. The manuscript would benefit from a more detailed discussion on how the proposed approach advances the state of the art in yoga pose recognition and its potential implications for related fields.

While the underlying data is provided, the statistical analysis is somewhat basic and lacks depth. The robustness of the findings could be further substantiated by employing more advanced statistical methods to analyze the results, such as confidence intervals or hypothesis testing, to better quantify the significance of the reported accuracy improvements. Additionally, while the conclusions are generally well-stated and linked to the research question, they occasionally overreach the data presented, particularly in claiming superiority over all existing methods without sufficient comparative analysis. It is recommended that the authors temper their conclusions to align more closely with the evidence provided, ensuring that all claims are fully supported by the data.

Cite this review as

Reviewer 2 ·

Basic reporting

May display the snapshot of extracted global feature and local feature in the manuscript
The feature fusion function may please be discussed in detail with associated algorithm
Detailed motivation behind using dual structure convolutional neural network may be discussed in the manuscript

Experimental design

For line no's 137 to 138 and other places why font size of dimensions (50x50x16) has been kept different "we consider a double layer structure, including convolution layer, batch normalization 50x50x16... "
Snapshot of the dataset may be provided in the mansucript
Why only ReLU has been choosen. Justify it
May add motivation section before contributions
Add a separate discussion section before conclusion
English needs improvements
If possible, may add latest refrences 2023,2024

Validity of the findings

How are the obtained results different?
Comparative analysis needs to be highlighted.
The applicability of the study may be discussed

Cite this review as

Reviewer 3 ·

Basic reporting

The English grammar should be polished in the revised version.
The figure quality is very bad. it should be improved in the revised version.
Extend introduction section by adding latest updates.
The explanation of each equation should be mentioned, currently mostly equations lack it.

Experimental design

How are the parameters used for algorithms determined? Has a Preliminary Study been conducted? Or do they take from another paper? They need to be more clear about that.
The methodology must be explained in a more detailed way to help the reader understand the parallel recognition approach in the given study.
Several new citations and references from recent research articles should be included in the manuscript.

Validity of the findings

The novelty of research by the authors should be proved by a comparative study.
Highlight the applications and utility of the work.
The authors should add a discussion section.
Please mention the key findings of your study.

Cite this review as

---

## Round 0.2 · accepted · Accept

Dear authors, we are pleased to verify that you have met the reviewer's valuable feedback to improve your research.

Thank you for considering PeerJ Computer Science and submitting your work.

Kind regards
PCoelho

Reviewer 2 ·

Basic reporting

Comments have been adressed

Experimental design

no comment

Validity of the findings

no comment

Additional comments

no comment

Cite this review as

Reviewer 3 ·

Basic reporting

I enjoyed reading this manuscript and believe that it is very promising.
Overall, this is an interesting study, and the results obtained are good.

Experimental design

The authors have improved the manuscript as per my previous suggestions. No new suggestions. The manuscript can be considered for acceptance.

Validity of the findings

This method is reliable, rapid, and useful for prediction. There are no new suggestions. The manuscript can be considered for acceptance.

Additional comments

The manuscript has been revised as per my comments and suggestions.

Cite this review as